# Emotivism Trends in Digital Political Communication: The Influence on the Results of the US Presidential Elections

Belén Casas-Mas [1],*, Martin Fernández Marcellán [2], José Manuel Robles [3] and Daniel Vélez [4]

1 Faculty of Media & Communication Science, Universidad Complutense de Madrid, 28040 Madrid, Spain
2 Faculty of Mathematical Science, Universidad Complutense de Madrid, 28040 Madrid, Spain; martif12@ucm.es
3 Faculty of Political and Social Science, Universidad Complutense de Madrid, 28040 Madrid, Spain; jmrobles@ucm.es
4 Department of Statistics and Operations Research, Universidad Complutense de Madrid, 28040 Madrid, Spain; danielvelezserrano@mat.ucm.es
* Correspondence: bcasas@ucm.es; Tel.: +34-91394-2143

**Abstract:** Citizens, organizations and institutions are increasingly making use of digital social networks such as Twitter as a means by which to express their position as regards political topics. However, an increasing amount of academic literature coincides, in that it highlights the emotive and expressive nature of these positions. In other words, for the most part, the political opinions that are publicized are more like backing based on support or rejection (without arguments or motives). In parallel, said expressions have a key emotional element (expressions of a positive or negative affective disposition). This article consists of an analysis that aims, on the one hand, to measure the affective disposition of US citizens as expressed on Twitter during the two most recent electoral campaigns (2016 and 2020). Similarly, we have generated a model that facilitates the measurement of the extent to which the content of the aforementioned messages demonstrates arguments or motives, or lack of. By way of the use of a database for the same Twitter accounts in both elections, we provide very consistent results that highlight the lack of argumentation of the public debate and the notably polarized profile of the affective dispositions shown by participants. We use the thesis of emotivism to give a more significant analytical weighting to this research.

**Keywords:** emotivism; social network analysis; sentiment analysis; political communication; digital communication; US presidential elections

## 1. Introduction

Over the last 15 years, the internet in general and, more specifically, social networks such as Twitter, Facebook, Instagram and TikTok have become a space for public debate. This is of particular relevance in terms of digital debates that arise during electoral processes. The form acquired by said debate on social networks (open, horizontal, polarized, fragmented, etc.), in addition to the content of the interventions made by citizens, media and representatives, informs us regarding the social construction of the political discourse, in addition to the way in which we define objectives, topics and motivations that converge for the election of one candidate or another. The study of this type of processes is therefore key in terms of an understanding regarding the sociopolitical reality of highly digitalized societies.

In recent years, several authors have warned of certain digital debate processes. Whether as the result of the network structure itself, or as a consequence of the way in which the political debate has been posed (marked by ideologies such as populism or right-wing radicalism), these processes are generating a context of digital political debate defined by expressions of support with no argumentative justification, in addition to being of a notable emotional nature. The first of these processes, the expressions of support for a

candidate that are not backed up by clear arguments, has led some authors (Robles and Córdoba 2019), to coin the term "participation based on Boos and Hurrahs". In other words, a form of participation in which only unconditional support or rejection is expressed. The emotive tendency either for or against a candidate is what determines the direction of these types of expressions. That is to say, for would be a "hurrah" (positive) and against would be a "boo" (negative).

This circumstance has consequently led to the generation of an environment tainted by a certain pessimism, as regards the political possibilities of the internet. We have therefore moved from a context (of "digital democracy" and "mass self-communication") that promoted a process of political revitalization as a result of its digitalization, horizontalization and inclusion of those traditionally excluded, to a situation in which we observe processes such as incivility and polarization.

The electoral processes in the US during the recent decades have been the core center of many studies between 2016 and 2019, because of the consequences on the nature of communication due to the transfer of political debate to digital networks. Among these studies, it is worth mentioning those that have addressed the increase in the trend of feelings, both positive and negative, among presidential candidates (Gelman et al. 2021; Losada et al. 2022; Evans et al. 2022). In 2023, the United States led the world in the number of Twitter users, with a base of 64.9 million. North America is the region with the highest number of Twitter users of all regions (Statista 2023)[1].

This article aims to measure and evaluate this process throughout two electoral campaigns in the US (2016 and 2020). In order to do so, we have taken a database consisting of messages shared on Twitter (since July 2023 it has been renamed "X") by the same accounts during both elections. This enables us to generate a more robust and accurate comparison between both elections (to observe whether there was movement in one direction or another). This facilitates the analysis of the percentage of messages that contain contended debate proposals (from now on we call them "messages with content"; e.g., Trump is sexist because he denigrates women in his speeches!) compared to those that are fundamental expressions of support (e.g., Long live Trump!) or rejection (e.g., Retire now, President!) ("messages without content"). Likewise, we have undertaken a sentiment analysis to understand the direction of these messages (for or "hurrahs" and against or "boos").

This article will therefore proceed as follows. We will first summarize the theoretical references that inform this study. Section 2 of this article will then go on to describe our database, the methodology employed, and the main results obtained. Lastly, the discussion section will consist of a reflection regarding the results, in addition to an assessment that will contextualize our findings.

## 2. Theoretical Framework

During the early years of the internet, and particularly following the boom of Web 2.0, there appeared to be a particular academic consensus as regards the potentialities of the internet tools as new means of channeling public opinion (Castells 2012; Stier et al. 2017). The known model of "mass self-communication" (Castells 2012) was considered to be a system in which any user has sufficient potential to modify, support, mobilize or influence political and social conduct without having to be a present or authoritative participant in political processes (Castells 2012; Dader and Campos-Domínguez 2017).

In accordance with this perspective, social networks would permit a type of participation based on collective authority and multidirectional communication (Serrano-Contreras et al. 2020). A significant amount of academic research focused on the possibilities of the internet and the online environment heading towards a "communicative ideal", first proposed by researchers such as Habermas (2006). On several occasions, this phenomenon was explained on the grounds that network tools facilitated both integration and the acquisition of the knowledge required to participate in politics (Kahne and Bowyer 2018). Thus, we would find ourselves in a context in which citizens would be able to confront

the communicative, political, economic and social monopoly that traditionally falls to conventional socio-political agents (Benkler 2006; Castells 2012).

In this sense, studies began to emerge in support of the optimistic discourse. At the beginning of the 21st century, most experts considered the internet and social networks to be the "awakener" of political participation, in addition to being the ultimate legitimizer of the representative democratic model (Robles and Córdoba 2019). Some authors considered that the Internet could improve representative democracy by strengthening the links between representatives and represented, offering more information to citizens about political processes (Hacker and van Dijk 2000).

This thesis, which is far from disappearing, continues to serve as a basis for more recent work. These studies demonstrate a positive correlation between the use of these network tools and participatory engagement (Boulianne and Theocharis 2020), emphasizing the role played by social networks, media, blogs, or democratic tools provided by institutions.

Other authors have found evidence that points to disintermediation dynamics. Under this approach, the aim is to reduce content production costs along with the ease of distributing them, as the main communication effect of Web 2.0 technologies. In this process of disintermediation (Benkler 2006), the Internet user can produce content and make it reach a global audience, limiting the action of traditional mediators. The inclusion of citizens who are not part of these organizations entails horizontality in communicative relations (Surman and Reilly 2005).

There is another theoretical axis that encompasses the theory of equalization or leveling (Ward and Gibson 2009; Yang and Kim 2017). This approach maintains that the current communication context is hybrid because it includes both traditional and new mediators in the public space, the latter named by Benkler (2006) as "amateur citizens". In this comparable process, traditional communicative culture (whose operation depends on structured and highly hierarchical organizations) and digital culture (more horizontal and based on connective action) converge (Bernroider et al. 2022; Shahin and Ng 2022). The most optimistic authors assess that partisan exposure to social media promotes political participation and exposure different political views (Mutz 2002). Furthermore, social media can contribute to incidental news exposure (Kümpel 2019) and increasing exposure diversity (Bodó et al. 2019).

On the contrary, the cyber-pessimist authors support the idea that the real structure of the online debate is in fact far removed from the standards imagined by the first social researchers. The transformative factor attributed to the network is, from this new perspective, merely a mirage. According to these studies, the role of political actors has not changed following the appearance of digital social networks, meaning there is a certain "normalization" effect. That is, the transferal of "normal" power relations from the offline space to the online space (Margolis and Resnick 2000; McGregor et al. 2017). Therefore, both the collective and individual potential that early research grants to users, as opposed to classic actors, would come as a result of the optimism awakened by paradigm shifts (Margolis and Resnick 2000; McGregor et al. 2017), although in practice these standards are far from reality.

The most widespread studies regarding the obstacles to public discussion on the internet refer to the structural issues of the digital tools themselves. Some researchers highlight that social networks create opportunities for filtering information and contacts and interactions, favoring processes of both entrenchment and polarization. Echo chambers refer to the same process by observing the ways in which users tend to cluster around voices with similar ideologies (Shore et al. 2016). Consequently, social networks enhance a discourse far from the ideal of the public debate.

In the same vein, warnings have been made as regards the impact of artificial intelligence tools on digital public debate. The boom of predictive algorithms and the micro-segmentation of audiences have encouraged the launch of bots that mimic human behavior with the purpose, in some cases, of sabotaging the debate and channeling the con-

versation to take advantage of the public positioning for or against certain issues (Bradshaw and Howard 2018; Martínez Torralba et al. 2023).

Secondly, together with this structural perspective, we can observe a second focus that questions cyber-optimist theses. Specifically, this second perspective refers to the limitations of attitudes and dispositions (Papacharissi 2015). This article focuses on this second instance in order to advance knowledge regarding the limits linked to processes of argumented expression of political ideas and opinions shared on social networks. Likewise, this limitation would be accompanied by an affective disposition that gives non-argumentative (without content) expressions with a certain emotional tone.

Finally, the main objective of this study is to analyze to what extent the messages issued during the debate process are or are not based on arguments and whether said arguments are positive (hoorays) or negative (boos).

### 2.1. Emotivism

To give more theoretical and analytical weighting to our proposal, the study of the limitations in the presentation of political opinions expressed on social networks, we employ the use of the concept of *emotivism*. This is a resource of metaethical current that proposes the idea that emotional expressions do not affirm anything about a given issue, i.e., they are neither true nor false. They are statements regarding our own individual sentiments. In this case, there is no need to argue, but instead seek affective attunement with expressions that make others feel a certain way. From this point of view, the increasing reduction of argumentative expressions and the pre-eminence of the presence of affective dispositions is more understandable.

*Emotivism* claims that the pillar of moral experience is not found in reason, but in emotion. Contrary to Habermas' cognitivist *Theory of Communicative Action*, emotivism rejects the existence of objective matters open to reasoning on which citizens can agree (Stevenson 1950). It also rejects naturalistic theories that place empirical evidence that something is better or worse than something else at the center of debate and decision-making. It essentially establishes that moral judgments are irretrievably linked to sentiments and states of mind, as opposed to objective or reasoned facts, as stated by Hume in *A Treatise of Human Nature* (1738).

There are, however, empirical studies that demonstrate the impossibility of achieving a discussion on social networks in which participants move away from their previous notions and opinions, despite the new information they receive. Faced with the cognitive effort that participation in a debate of socio-political significance can entail, users are more influenced by their initial claims than by other new arguments (Taber and Lodge 2006; Garrett and Stroud 2014; Prior 2013), thus questioning the central basis of the public debate idea.

Related to the role of emotions on social media political communication, there are studies that have shown the positive and negative emotions used by political parties to increase popularity cues that trigger reactions such as likes in Facebook (Blassnig et al. 2021). This platform is crucial for designing political campaigns where candidates aim to mobilize their followers through optimism, or to promote negative emotions towards other candidates or parties (Eberl et al. 2017). Experimental research on the effects of populist communication have demonstrated that messages of anti-elitism or excluding out-groups and negative portrayals of other candidates increase "angry" reactions; on the contrary, the anger is reduced when the populist discourse is inclusive or when citizens participate with optimistic or positive messages (Jost et al. 2020).

With this in mind, we start a conversation regarding discussions on social networks that revolve around emotive expressions against or for an issue, which we will refer to as "boos" and "hurrahs". This dichotomous approach between positive and negative represents purely affective dispositions that entail both the reinforcement of a position and its social sanction.

*2.2. Structures of Emotivist Expressions*

Emotivist expressions can be structured based on three different categories that entail different forms of communication: expressions of sentiment; imperative expressions; and expressions for or against a certain object. We will proceed to give some examples of their structure based on messages from previous research.

Expressions of sentiment are those that consist of messages of approval or rejection of a specific fact or cause. When user A uses the sentence "with you all the way, Pablo!!!", what they are really trying to say is something along the lines of "I approve of Pablo's proposal/action". Another user B who reads this message can either share what user A has stated, or not. However, especially if user B disagrees with what user A has said, the debate would be nullified as there is simply nothing to debate about, beyond the obvious fact that user A is stating their sentiments on the matter.

While it may be true that user B could ask user A for the reasons behind their approval of the position taken by Pablo and establish a sort of Socratic debate that would reveal the deeper reasons for their support, this is unlikely, as there is no record of such debates taking place on social networks. This study accepts the idea that the behavioral and expressive behavior of the social network users tends towards emotivist positions. In other words, we adopt a methodological emotivist approach. It is therefore important to emphasize that the conversation centers upon the motives with which they agree.

That is, the participants in the debate generally require satisfactory reasons in order to be able to have sufficient influence on their prior notions. However, these reasons are not given in this type of emotivist debate. In the present study, we do not assess that people are emotivist but behave as such in social media debates.

The second type of expression goes beyond the mere disclosure of sentiments. The main purpose of imperative expressions is to challenge the reader. An example might be "this journalist should just shut up, for God's sake!!!". Once again, debate is unlikely to arise as a result of this type of statement, as they do not even raise the issue. They do not incorporate any useful information or any resources to encourage conversation. Its purpose is to find sympathy or affinity with the other users. In other words, it aims to appeal to like-minded users without resorting to reasoned reasons for agreeing with said position.

Lastly, the third type of emotivist expression has been subjected to criticism from the main critics of emotivism. We are referring to expressions for or against attitudes, facts, or actions. These statements are considered to be of the type "Political corruption is the worst of the evils of democracy" and are frequently used as arguments to expose or present one's own judgments regarding controversial issues. In this research, such expressions will also be referred to as "boos and hurrahs". In this sense, the main criticism of emotivism is that expressions for or against leave a door open for discussion or debate. In this case, a recipient A could reply with an expression of reasoned support ("in addition, corruption is a bad example for citizens") or reasoned disagreement ("corruption is inevitable because it is at the basis of politics"). In this case, a critic would argue that reasoned discussion is possible from an emotivist perspective. However, the aim of such expressions for or against is not to encourage debate, but to generate influence by trying to change or override the sentiments or preconceived notions behind other people's judgments. They seek to generate a sentiment—that sentiment may be sympathy, empathy, fear, anger, etc.— that justifies the reader's support or rejection of a given issue by appealing to shared internal values.

In this paper we have tried to operationalize this concept and apply it to a real case. As mentioned earlier, the aim is to quantify the presence of messages that contain arguments compared to those that are, in essence, emotive expressions. To find out the emotive direction of emotivist messages, we enrich the study with an analysis of emotions (for or against the candidate).

## 3. Materials and Methods

In this section we will first summarize the technical details of the database used for this article. Secondly, the techniques and methods used will be presented, and finally,

the main results that will serve as the basis for the discussion section with which we will conclude the article.

### 3.1. Data Collection

The tweets included in the database were collected throughout the two months prior to the US presidential election campaigns held on 8 November 2016 and 3 November 2020. The data were collected using the API that Twitter provides to its users. The tweet capture processes for both elections resulted in a total of 4,076,176 tweets for 2016 and 31,367,552 tweets for 2020. The tweets captured form part of conversations in which the username of one of the candidates running in the elections has been mentioned. In other words, the tweets that have been analyzed as part of this project directly mention one of the candidates or form part of the replies to tweets that make said mention. Messages referring to Donald Trump or Hillary Clinton (2016 electoral campaign) and Donald Trump or Joe Biden were captured (2020 electoral campaign).

Given that the aim of this research is to undertake a comparison of the messages that refer to Donald Trump in 2016 with those in 2020, we selected tweets that made exclusive allusions to the other two candidates. To make a reliable comparison, we decided to undertake the exclusive analysis of tweets from users that appeared in both databases (Trump, 2016 Elections and Trump, 2020 elections databases), in order to appreciate the changes in both the expressions and demands of citizens, organizations and institutions as regards Trump. As a result of this, our two databases accounted for a total of 1,041,410 messages for 2016 and 1,477,300 for 2020. We therefore worked with a total of 2,518,710 tweets.

The tweets from both databases will be analyzed using two variables: sentiment and content. The sentiment of a tweet consists of determining whether the tweet demonstrates a positive, negative, or neutral opinion of Donald Trump. On the other hand, the content analysis consists of determining whether the tweet presents an argument or whether it is a fundamental expression of support or rejection ("boo" or "hurrah").

Supervised classification techniques were applied and given the double objective (sentiment and content); each tweet was tagged on two variables. A model was therefore used for each of the two variables. To train these models, a sufficient number of tweets had to be manually tagged in order to create a sufficiently reliable model. In this case, we decided to tag 3000 tweets, half of which correspond to each year.

As regards the sentiment tagging criterion, we made the logical decision that if the messages supported Trump they were tagged as positive, whereas those that demonstrated criticism were tagged as negative. On the other hand, if they disqualified Donald Trump's rival candidate, the messages were considered to be positive. Meanwhile, if they praised the rival candidate, they were considered negative. Tweets from news agencies reporting updates on the campaign would be considered neutral.

### 3.2. Methodology

As mentioned earlier, two models were created to tag the tweets on the two axes (sentiment and content), to identify whether the tweet has content and if the tweet is positive or negative. Following the labeling of the 3000 tweets, it can be observed (Figure 1) that:

- In the tagged tweets from 2016, the sentiments were quite balanced: 36.4% of the tweets were positive, 34.5% were negative and the remaining 29.2% were neutral. However, tweets without content were clearly in the majority, registering 69.3% of tweets, compared to 30.7% that do have content. It could be seen that the vast majority of negative and neutral tweets had no content.
- In the tagged tweets from 2020, negative sentiments represented a considerable majority (44.5%), while the positive sentiments remained at a similar percentage to that of the tagged tweets from 2016: 34.6%. The percentage of neutral tweets dropped drastically to 20.9%. In terms of content, there were also differences with the tagged tweets from 2016. Tweets with content accounted for 40.9% of the total and tweets without content accounted for 59.1%. It could be seen that only in neutral tweets

was there a considerable difference between tweets with content and tweets without content.

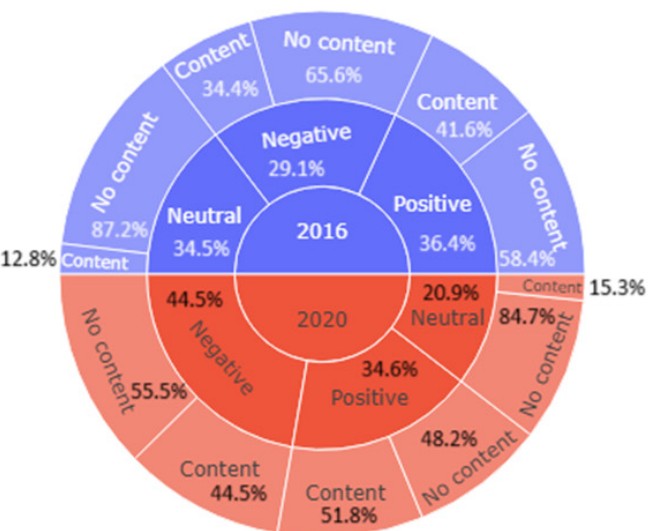

**Figure 1.** Statistics of the 3000 manually tagged tweets.

Subsequently, as a prior step to adjusting both models, the textual information was processed using NLP (natural language processing) techniques (it was sourced by the authors using the resources to the Complutense University of Madrid, Spain) that allowed it to be converted into a table that contained as many records as tweets and as many columns as terms (tokens)[2].

In the resulting table, each pair (tweet, token) was associated with its tf–idf value (term frequency–inverse document frequency), a value that allows us to reflect the importance of each token in each tweet, considering the set made up of all the tweets. Subsequently, the table was partitioned into two: a sample with 2000 tweets that served as training for the models and another with 1000 tweets that served as a test sample and that allowed the validating of their generation capacity. In both groups, half of the group are tweets from 2016 and the other half from 2020.

Regarding the typology of the contrasted models, in both cases decision trees, logistic regression, linear and quadratic SVM (Support Vector Machines), random forest and gradient boosting were tested.

### 3.2.1. Sentiment Analysis Model

To adjust the sentiment analysis model, tweets labeled with neutral sentiment were eliminated, considering that the subjectivity of said label could introduce noise into the training process. This restriction meant reducing the dimensions of the training and test tables to 1574 and 676 tweets, respectively.

This model proposed predicting/characterizing the "positive" label based on the tf–idf values associated with the tokens. The winning model in terms of the success rate on the test table was a linear SVM.

The model is correct in 512 of the 676 tweets that make up the test table, providing an accuracy rate of 75.74% (Figure 2).

Of the 344 times that the model predicts a negative sentiment, it is correct 276 times, which translates into a precision in estimating said sentiment of 80.23%. On the other hand, of the 331 times that the model predicts a positive sentiment, it is correct 236 times, which translates into a precision in estimating said sentiment of 71.30%. In this way, it can be seen that the model is less wrong in predicting negative feelings than positive ones.

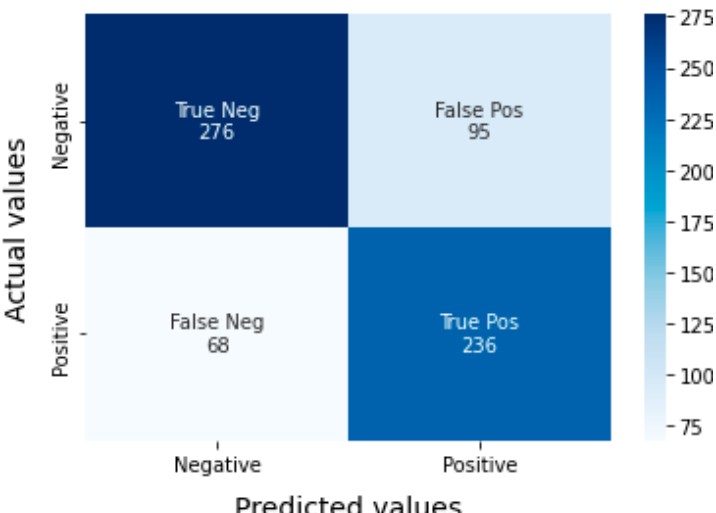

**Figure 2.** Confusion matrix for the sentiment model.

### 3.2.2. Content Model

To adjust the content model, tweets with hashtags were eliminated, considering that if a tweet has a hashtag, then it has content. After filtering these tweets, the training and test tables were restricted to 1458 and 748 tweets, respectively.

This model proposed predicting/characterizing the label "with content" (which determined whether a tweet contained some type of argument) and the label "without content" (which determined that it was essentially a message consisting of an insult or compliment), depending on a list of inputs generated for this purpose:

- Percentage of emojis that the text has. To do this, the number of emojis present in the tweet is counted and divided by the number of words that the original tweet has. A priori, the use of emojis should be an indicator of lack of content, but many news agencies use them to attract attention, so their influence may vary depending on the context.

- Number of connectors in the tweet. The connectors can be of various types (see Table 1). It is to be expected that the more connectors a tweet has, the more likely it is to be tagged as a tweet with context.

- Usage percentage. A priori, the longer the message, the more likely it is to have content. In this case, the length of the message in characters will not be measured, as between the 2016 and 2020 elections, specifically in 2018, Twitter decided to double the maximum length of tweets from 140 characters to 280. This therefore meant that the lengths would not be comparable and there would ultimately be a tendency to think that the 2020 tweets are more likely to have content as they have more characters. To address this issue, we decided to calculate the ratio between the number of characters used by the user and the number of characters available. The step prior to calculating the percentage of use is the filtering out of elements that form part of the context of a conversation or questioning, such as mentions of other users, which do not contribute to the message and use up characters. Therefore, they should not form part of either of the two characters counts that are made, neither characters used nor characters that could have been used. This is also applicable to links, as they do not form part of the user's original message but take up a significant number of characters and will therefore not be counted either. Then the percentage of use is calculated by employing the use of Equation (1):

$$\text{Usage percetage} = \frac{l_{post}}{l_{max} - (l_{pre} - l_{post})} \tag{1}$$

where $l_{max}$ is the the maximum length allowed by Tiwitter at the time it was written, $l_{pre}$ is the length of the tweet prior to filtering and $l_{post}$ is the length after filtering. Note that if $l_{post}$ value is equal to $l_{max}$ value then $l_{pre}$ value is equal to $l_{post}$ value (because $l_{pre} \geq l_{post}$) and the ratio is equal to one. On the other hand, the value zero of the ratio is associated with the value zero of $l_{post}$.

- Complexity of the message. In principle, the more complex the message, the more likely it is to be a message with content. To measure the complexity of the message, the Flesch–Kincaid readability test (1948) (Flesch 1948) has been used, which aims to measure the ease of understanding a text in English. To do this, we rated the texts with a score from 0 to 100, with a text with a score of 0 being a very difficult text to understand and a text with a score of 100 being understandable by an 11-year-old child. To obtain this value, we followed Equation (2):

$$\text{COMPLEXITY} = 206,835 - 1.015 \left( \frac{\text{number\_of\_words}}{\text{number\_of\_sentences}} \right) \\ - 84.6 \left( \frac{\text{number\_of\_syllables}}{\text{number\_of\_words}} \right) \tag{2}$$

As is the case with the percentage of use, processing is applied to the texts prior to calculating their complexity index.

**Table 1.** Connectors used in the project, classified by type.

| Connector Type | Connectors Used |
|---|---|
| Additional information | Also, in addition, moreover, additionally, and furthermore. |
| Nuance | In fact, precisely, actually, at the far end and justly. |
| Exception | Unless, if, as long as, though, although and even so. |
| Concession | However, in any case, anyway, yet, even so, although. |
| Cause | Because, due to, through, therefore, since, as and considering. |
| Consequence | Consequently, therefore, so, thus, and hence. |
| Clarification | That is, in other words, namely and more clearly. |
| Comparison | Compared to, likewise, similarity, in the same way and equally. |
| Temporality | After, before, now, next, while, at the same time, later, currently, formerly, and subsequently. |
| Intention | With the purpose of, for and for the purpose of. |
| Adversity | On the contrary, by contrast, on the other hand and instead. [1] |

[1] Resource: Own elaboration based on the tweet's manual classification.

In this case, the processing applied consisted of eliminating links, eliminating usernames and separating the hashtags into the different words that compose them.

Of all the models tested, it was the decision tree that provided a better misclassification rate on the test table.

- In said tree, the maximum number of branches was set at three, the maximum depth at three and the minimum number of observations per leaf at fifteen observations, which corresponded to 1% of the training sample. Finally, it should be noted that the pruning of the tree was carried out using a cross-validation strategy using the mean square error. The resulting tree is the one presented in Figure 3.

The first variable that participates is complexity, its influence being the expected one: the lower its value, the more complex the tweet will be and, therefore, the greater the probability that it is a tweet with content.

The variables that participate in the second level of depth of the tree are the number of connectors: the more connectors and the percentage of use, both responding to the expected logic. Regarding the first, it is observed that the greater the number of connectors, the greater the probability that the text will have sufficient arguments to consider it as a "tweet with content". Regarding the second, the greater the percentage of characters used in writing (over the maximum possible), the greater the probability that the text has content. Finally, in the last level of the tree, the variable percentage of emojis participates.

In this case, the higher the percentage of emoji, the greater the probability that the tweet has content, which a priori contradicts the expected logic. However, we consider that this may be due to the extensive use of emojis by news agencies or simply because normal users use them to give more content to their message by accompanying them with the appropriate emoji. In any case, the conclusion associated with this variable must be interpreted with caution given that, as can be seen, there are only 30 tweets that support one of the nodes.

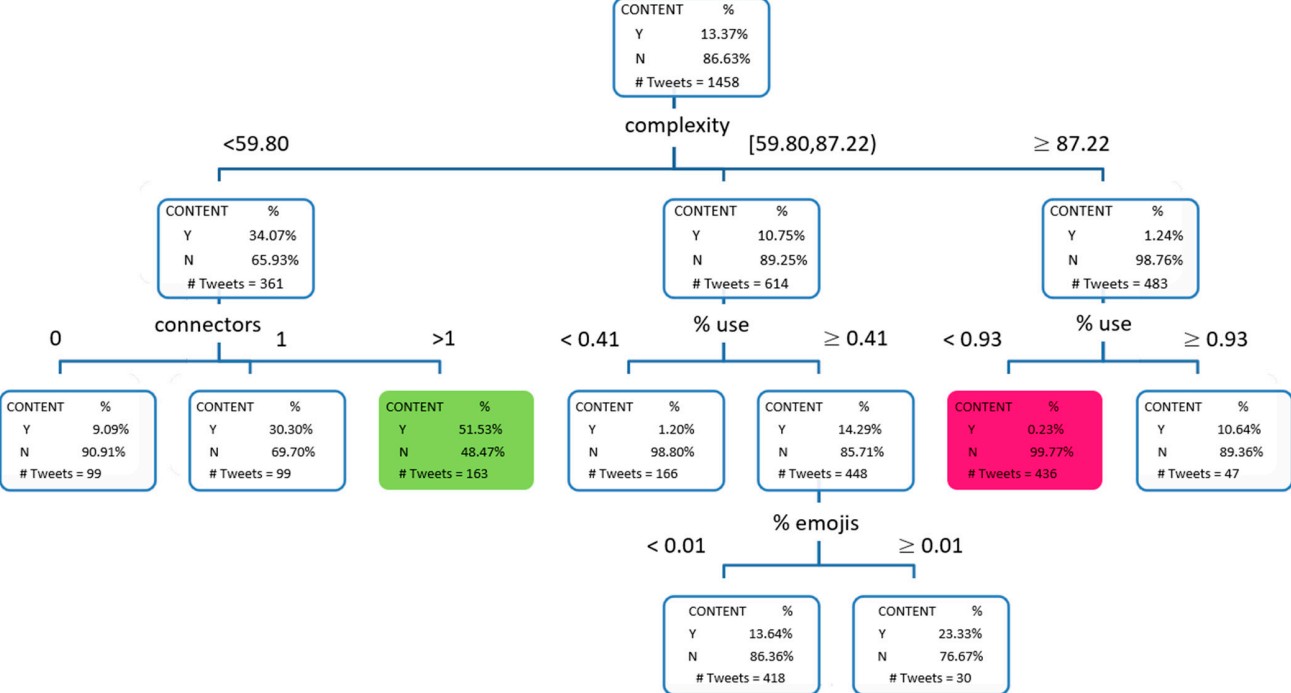

**Figure 3.** Graphic representation of the adjusted decision tree.

Finally, we can conclude that the profile to which a tweet without content is most likely to respond is the one with a low degree of complexity and a length that uses less than 92% of allowed characters (pink node in Figure 3). Of the tweets that respond to this profile, 99.7% have no content. A tweet that obeys this pattern would be as close as possible to a Boo if it had been labeled as negative by the previous model and as close as possible to a Hooray if it had been labeled as positive.

On the other hand, the tweets that present more content are those of low complexity and with at least two connectors. Of the tweets that respond to this profile, 51.53% do have content (green node in Figure 3), multiplying the probability that a tweet has content almost by four (13.37% as can be seen in the initial node of the tree).

Regarding the model's success rate on the test table, the confusion matrix provided the following results (Figure 4).

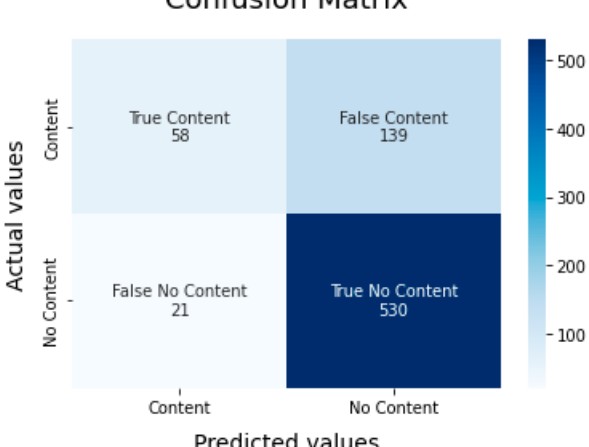

**Figure 4.** Confusion matrix of the content model.

## 4. Results

Once the models were adjusted, they were applied to the total number of tweets available in each of the electoral periods, thus assigning to each tweet the probability of having positive sentiment and having content. In both cases, the value 0.5 was established as the probabilistic threshold to consider that a tweet has positive sentiment and has content. The results derived after this assignment are presented below. These results have been translated into a list of pie charts in which, for each electoral period, the proportions of tweets that respond to one or another label are displayed. Associated with each of them, a test has been carried out to contrast whether the difference of said proportions can be considered equal to 0, using for this purpose the following statistic:

$$Z = \frac{p_{16} - p_{20}}{\sqrt{\frac{p_{16}*(1-p_{16})}{n_{16}} + \frac{p_{20}*(1-p_{20})}{n_{20}}}}$$

where $p_{16}$ and $p_{20}$ are the proportions observed with respect to the label of interest in the 2016 and 2020 electoral periods, and $n_{16}$ and $n_{20}$ are the number of tweets labeled by the model that generates said labels. If the value of this statistic is, in absolute value, greater than 1.96, it can be considered that the difference in proportions is significant, at a significance level of 0.05.

The fact that the sample sizes used are so high leads us to reject all the tests. So, the proportions involved in them can be considered significatively different.

To start with, in terms of sentiment analysis, the percentage of positive tweets is significantly greater in 2016 than in 2020. Figure 5 shows how opinions about Trump published on Twitter go from having a clear positive majority of 61.5% to positive opinions being a minority, in just 48.4% of tweets.

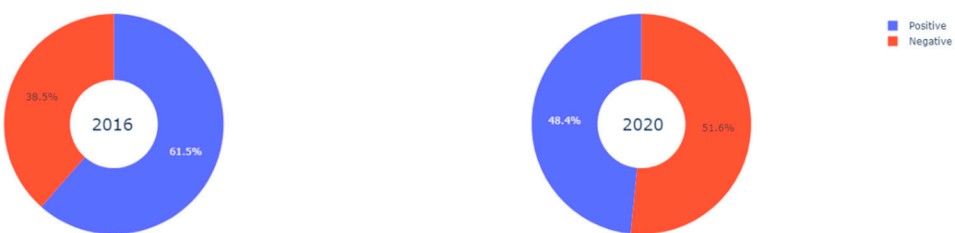

**Figure 5.** Distribution of sentiment of tweets per year.

This reduction in positive opinions could be due to the fact that a number of the Trump voters changed their opinion from 2016 to 2020 or, essentially, the "attrition" effect of the

government management itself. It is also possible that people who did not vote for Trump became more active Twitter users, to prevent a second Trump victory.

In terms of content analysis, the percentage of tweets with content and without content is very similar in both elections, as shown in Figure 6. Thus, despite the fact that there has been a loss of support, in emotive terms, for President Trump, the way in which this information has been presented (in terms of argumentation or not) is very similar.

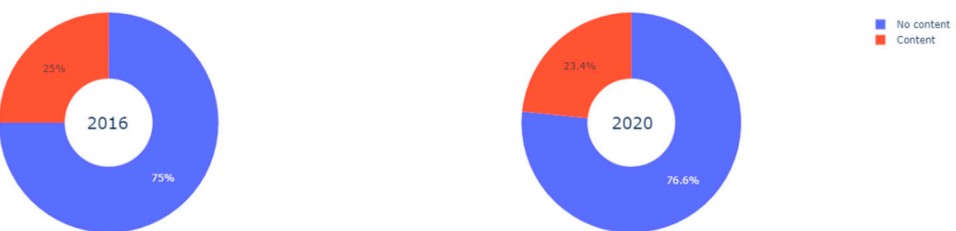

**Figure 6.** Distribution of tweets according to content by year.

It seems clear that the hegemonic form of expression on Twitter during the two campaigns is by way of messages in which no arguments are expressed (messages without content). That means that we are facing a polarized electoral scenario in relation to the affective disposition of the participants involved in the Twitter debate and it is clearly marked by the support/rejection more than argument and public debate regarding ideas.

However, taking into consideration the fact that the tweets including a hashtag have been considered as tweets with content, it is reasonable to think that this is the case for most tweets with content, as they include hashtags, and not necessarily because they contain profound arguments. To verify this hypothesis, we extracted the number of tweets with and without hashtags for each year in search of relevant differences, in order to check whether the complexity of the messages was actually similar in both years.

Figure 7 shows that the number of tweets with hashtags is similar in both elections, but we can also observe that in both cases the number of tweets with hashtags is slightly lower than the number of tweets with content. Therefore, the majority of the tweets with content include hashtags. We can therefore see that, in 2020, there is a greater percentage difference between the tweets with hashtags and the tweets with content than in 2016. Consequently, we decided to filter the tweets with hashtags and calculate the percentages of tweets with and without content (Figure 8).

We can observe that the tweets with content and without hashtags are a significant minority; however, the percentage of tweets with content from 2020 is slightly larger.

Finally, it was decided to carry out an analysis of how the feeling and content models interact. To this end, two histograms (Figures 9 and 10) were first represented with the distribution of the probabilistic values assigned by the sentiment model to the tweets of each of the two electoral processes. In this way, the bars associated with the lowest probabilistic values respond to very negative feelings, those associated with higher values to very positive feelings, and the central ones to tweets with messages that could be classified as neutral. In turn, each of the bars represents the proportion of tweets that have been labeled with and without content by the content model. Thus, those tweets without content located in the left and right tails of said distributions would be those that we would identify as "booos" and "hurrahs", respectively. In view of these graphs we can draw some interesting conclusions:

- There are many tweets to which the sentiment model assigns probabilistic values close to 0.5, and which can therefore be classified as neutral, while the other model labels them as tweets without content. In this way, the neutrality of the message in terms of sentiment does not necessarily imply that it has content. As an example, one of the tweets that responds to this profile (neutral and without content) is: "@realDonaldTrump @victorpapalima how can we?".

- Of the most negative tweets (with probability between 0 and 10), in 2016 (Figure 9) the percentage of tweets labeled without content multiplies by 1.18 that of tweets with content (2.14/1.81), while in 2020 (Figure 10), this itself is multiplied by 1.58 (5.07/3.21). That is, we can identify more boos in 2020 than in 2016.
- Of the most positive tweets (with probability between 90 and 100), in 2016 (Figure 9) the percentage of tweets labeled without content multiplies by 0.28 that of tweets without content (3.06/10.75), while, in 2020 (Figure 10), this is multiplied by 0.56 (2.24/3.97). We can also conclude that there were also more hurrahs in 2020 than in 2016.

This being the case, we can observe how the messages of the 2020 electoral period seem to be more "emotivistic" than those of 2016 (Figures 9 and 10).

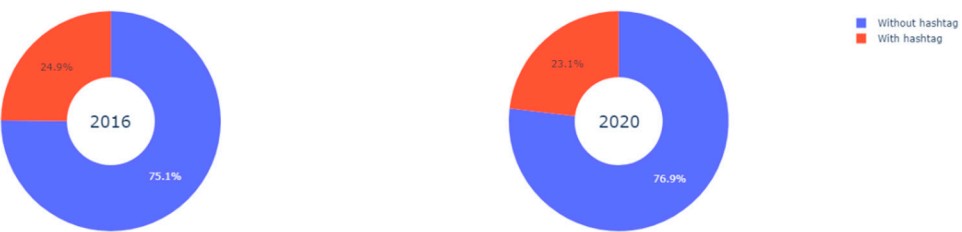

**Figure 7.** Distribution of tweets with or without hashtags by year.

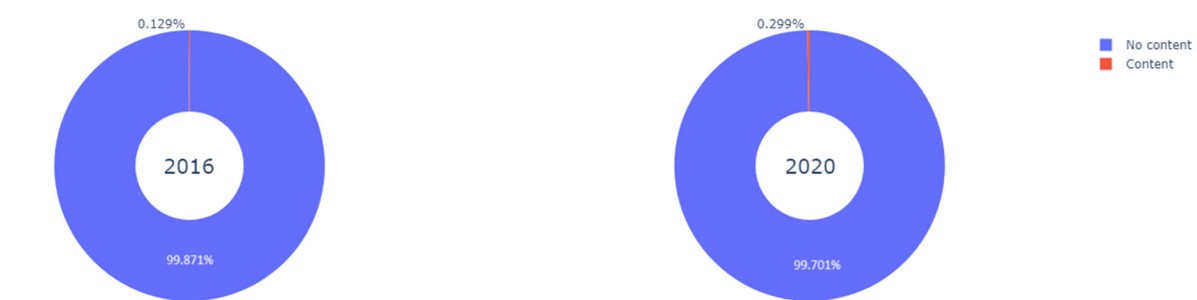

**Figure 8.** Distribution of tweets without hashtags according to content by year.

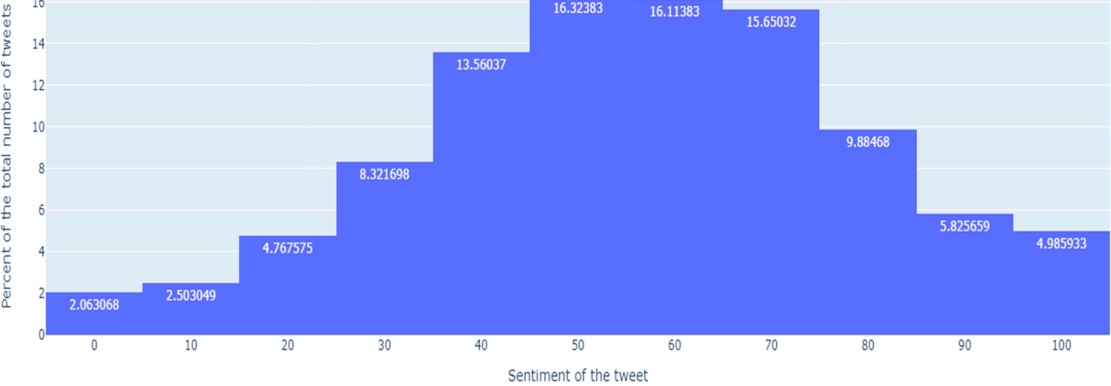

**Figure 9.** Distribution of sentiment analysis scores crossed with content labels, 2016 Elections.

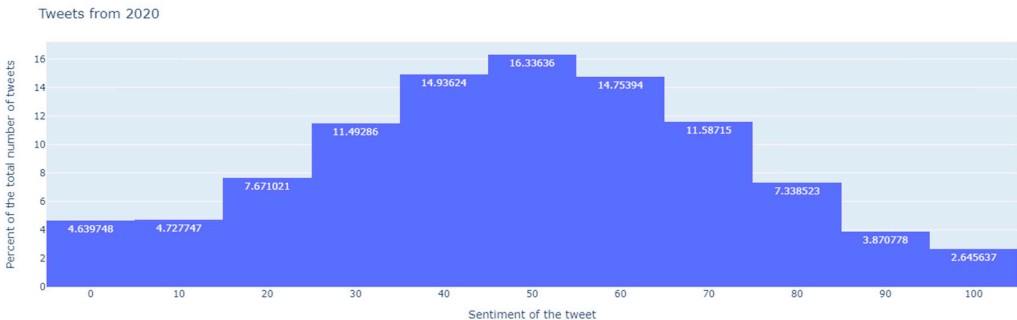

**Figure 10.** Distribution of sentiment analysis scores crossed with content labels, 2020 Elections.

## 5. Discussion

The aim of this research was to analyze the nature, argumentative or not, of the messages posted on the social network, Twitter, during the two most recent US electoral campaigns, in addition to their affective dispositions. As has been observed in the results section, the number of tweets with a positive affective disposition is practically the same in 2016 as in 2020. However, the most relevant finding of our study is related to its content. We found that approximately 75% of the messages posted in both campaigns were messages in which there was no evidence of argumentation (presumably based on expressions of either support or rejection). This percentage increased to almost 100%, as regards messages that did not include a hashtag.

This result should lead us to an initial conclusion; the overall tone of the debate studied is marked by messages in which the ideas expressed are not formulated in an argumentative format. This implies that the possibility of "understanding the reasons" of other individuals or groups tends to occupy a secondary position compared to other forms of expression. This finding points to the fact that proposals for the revitalization of the public space by way of the emergence of social networks and their politicization are not, at least in this instance, taking place on the basis of the ideal of debate public space. We speculate on the idea that such participation is produced, to a greater extent, by way of affective dispositions; that is, through an emotivist tendency, which goes in the same lines of the findings of other studies during the same electoral periods (Gelman et al. 2021; Losada et al. 2022; Evans et al. 2022).

However, we should distinguish between emotivism as a cultural tendency or trait and emotivism as an anthropological and philosophical theory. The former assumes that a methodological emotivism, though not necessarily, must be applied to the empirical and cultural context. The latter implies assuming a certain ontological and epistemological position. In our case, we do not affirm anything regarding the existence of an ontological emotivism, but we do affirm its factual presence in this type of debate.

Additionally, we emphasize the increasing negative nature of the messages that refer to President Trump. It is important to remember that these are the same accounts and that there is therefore an emotional transfer present in one in ten accounts. In other words, 10% of the accounts go from expressing themselves emotionally in support for Trump to expressing themselves emotionally against him. In contrast, the lack of arguments is consistent between the two years (75.1% and 76.9%, respectively, and 99.9% and 99.7%, respectively when referring to posts without hashtags). In other words, while Trump loses followers (affectively), potentially due to management attrition, both in 2016 and 2020 the ideas are expressed by the same accounts through messages that do not formulate arguments. This suggests a dynamic dimension of affective dispositions and a structural and more stable facet of message content.

Evidently, we come across empirical findings that reveal the unfeasible nature of achieving a discussion in the digital space in which network users distance themselves from their previous opinions despite the new informative inputs they receive. As participation in socio-political debates may require cognitive effort, internet users are less influenced

by new arguments than by their initial positions (Prior 2013), which therefore calls into question the core of the ideal of social debate on social networks.

From our point of view, the social conditions of communication are key to understanding the relationships people establish in the digital space. The sociohistorical framework that encompasses our reflection, coined "information society" by Castells (2011), offers a significant number of studies, according to which the use of emotionally charged speeches (Robles and Córdoba 2019) in which the debate focuses on expressions of sentiments of affection against or for certain people or issues. This dichotomous perspective (positive/negative, "boos/hurrahs") offers merely affective positionings that imply both the reinforcement of these and the social sanctioning of network users. What we cannot underestimate is the significance of that kind of effects when political parties develop their strategies and, more important, that those strategies might be addressed for political change (Mouffe 2022), which depends on the agents' interests and their alignment with those of society).

In short, our proposal from this study is based on the main idea that emotivism is an emerging cultural process in our societies that manifests itself in different phenomena and in different declinations that may or may not be related. It is the anthropological and philosophical approach in which emotivism operates, acting as a link of equivalence, thus implying that its trace has been detected in the electoral processes analyzed. This trace would not imply that the possible set of relationships or correlations has been exhausted.

At present, social media networks have brought a radical shift in the way citizens can access and disseminate information. This is largely an issue that political actors are using to build communities, in which emotions are the core of the debates. The implication of our study is that this phenomenon is reinforced by algorithms that shape public opinion, leading to the strong polarization of societies and a fatal blow to the quality of democracies. We consider that applying the methodology of our study to other electoral processes around the world would be very interesting, shedding light through comparisons of how emotions are being used biasedly to mobilize public opinion for exclusively partisan purposes. More research is needed to discuss the results and how they can be interpreted from the perspective of previous studies on political communication in digital debates. The findings and their implications should be discussed in the broadest context possible. Future research directions may also be highlighted.

**Author Contributions:** Conceptualization, J.M.R. and B.C.-M.; methodology, D.V. and M.F.M.; software, D.V. and M.F.M.; validation, D.V., M.F.M., J.M.R. and B.C.-M.; data curation, J.M.R., B.C.-M., D.V. and M.F.M.; writing—original draft preparation, J.M.R. and B.C.-M.; writing—review and editing, J.M.R. and B.C.-M.; supervision, B.C.-M.; project administration, J.M.R.; funding acquisition, J.M.R. All authors have read and agreed to the published version of the manuscript.

**Funding:** Research Group for Data Science and Soft Computing for Social Analytics and Decision Aid. This research was funded by national research projects funded by the Spanish Government (Ref. PID2019-106254RB-I00).

**Institutional Review Board Statement:** Not applicable.

**Informed Consent Statement:** Not applicable.

**Data Availability Statement:** The data used for this research was downloaded and analyzed without access to sensitive information or that compromises the privacy of any person or institution.

**Conflicts of Interest:** The authors declare no conflicts of interest.

## Notes

[1] Information available at: https://www.statista.com/statistics/242606/number-of-active-twitter-users-in-selected-countries (accessed on 3 January 2024).

[2] We should highlight that in the case of the content model, no stop words or lemmatization filters have been run, as it was considered that this could alter the final result, as some stop words may denote content.

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
