# Peer review of "Emotivism Trends in Digital Political Communication: The Influence on the Results of the US Presidential Elections"

_socsci, doi:10.3390/socsci13020080_

Round 1
Reviewer 1 Report
Comments and Suggestions for Authors
I have questions/observations for the authors:
1. the Twitter platform is currently called X. I realize that it was Twitter at the time of data acquisition, but it would be appropriate to clarify this point in the body of the article. In addition, the changes made by Elon Musk, have significantly affected the way X is used, perhaps including issues related to data retrieval and relationships. Every X user has experienced inconveniences related to the new algorithms and how posts are displayed on the feed. It is imperative to clarify this and indicate how the article author's findings relate to the current state - or are they only of archival value?
(2) Was it necessary to filter the downloaded data, in order to avoid noise, and eliminate fake accounts, bots etc., or was it not necessary? How was the authenticity of the accounts surveyed verified? What dictionary was used to assign sentiment to each word/phrase?
3 How did the authors deal with tweets containing errors and typos? How did they clean up the data in this regard?
4. please make your objectives and hypotheses clear and explicit. if the authors abandoned hypotheses in favour of an exploratory approach, please complete this clearly.
5. The theoretical foundation is not the strongest point of the article. It would be necessary (if only briefly) to refer to such concepts/concepts as mediatization of politics, politainment, post-truth (and more broadly:postmodernism). From the perspective of the cited names: Luhmann, Hepp, Denton&Woodward, McNair, Perloff, Renger&Wiesner.
Author Response
Dear Reviewer,
Thank you very much for your comments. We have tried to address the questions raised and we very much appreciate your time. Your review has helped us to improve our paper. We have enriched and/or clarified every section following your suggestions as indicated below.
1. The Twitter platform is currently called X. I realize that it was Twitter at the time of data acquisition, but it would be appropriate to clarify this point in the body of the article. In addition, the changes made by Elon Musk, have significantly affected the way X is used, perhaps including issues related to data retrieval and relationships. Every X user has experienced inconveniences related to the new algorithms and how posts are displayed on the feed. It is imperative to clarify this and indicate how the article author's findings relate to the current state - or are they only of archival value?
We have, accordingly, mentioned the current name of Twitter (L.52). Unfortunately, to relate our results with the present moment of the functionalities or analyzing problems of X, requires us to do comparative research. Thank for you comment.
Methods:
Was it necessary to filter the downloaded data, in order to avoid noise, and eliminate fake accounts, bots etc., or was it not necessary? How was the authenticity of the accounts surveyed verified? What dictionary was used to assign sentiment to each word/phrase?
When the data is downloaded from the leaders' official accounts, a filtering system is not necessary. We use a library of python to identify possible bots in our database and delete them
We have shown in the paper that the dictionary used to assign sentiment to each phrase is one called SentimentAnalysis implemented in Python, available in https://pypi.org/project/SentimentAnalysis/
3. How did the authors deal with tweets containing errors and typos? How did they clean up the data in this regard?
Thank you very much form this. In this case, nothing was done regarding typos and tweets containing errors because it is no necessary in this kind of dataset.
4. Please make your objectives and hypotheses clear and explicit. If the authors abandoned hypotheses in favour of an exploratory approach, please complete this clearly.
We have better defined our objective (Lines 129.-131) and, in the section of Discussion, we have explained how we reached it thanks to the methodology developed in this study.
Theoretical framework:
5. The theoretical foundation is not the strongest point of the article. It would be necessary (if only briefly) to refer to such concepts/concepts as mediatization of politics, politainment, post-truth (and more broadly:postmodernism). From the perspective of the cited names: Luhmann, Hepp, Denton&Woodward, McNair, Perloff, Renger&Wiesner.
In this article we address the theoretical concept of disintermediation (Robles and Córdoba, 2029), which is different from mediatization and poliaiment. That is the reason why we referred to authors that have previously developed that concept. We thank the references proposed by the reviewer but we consider that the concepts of post-truth, politaiment, mediatization of politics and postmodernity are too broad and general, which have little place in an empirical work like this.
On the other hand, we have eliminated the word “post-truth” (line.49) to avoid misunderstandings.
Yours faithfully.
Reviewer 2 Report
Comments and Suggestions for Authors
Emotions on social media are undoubtedly a relevant research field. Emotions make content on social media engage users and are therefore a driver of the distribution, visibility, and virality of content. When it comes to this paper, however, I have serious doubts about the quality of the study, both theoretically and methodologically, which is why I can’t recommend this paper for publication.
The authors state that they base their study on deliberative democratic theories. However, they seem to have a very superficial, reduced understanding of this approach. First, it is not the case that deliberative theories consider all kinds of emotions negative. Rather, they consider positive emotions positive since these can engage citizens and therewith lead to empowerment. Second, providing content (which is what the authors investigate in this study) is by far not enough from the perspective of deliberative theories. Rather, these have a strong focus on how the political discourse is characterized, if it for example fulfills criteria such as justification, civility, and neutrality. However, this is neither discussed in the theory section nor does the empirical study take any of these central criteria into account.
The authors‘ deficient understanding of democratic theories is underlined by the fact that they write: “At the 82 beginning of the 21st century, most experts considered the internet and social networks to 83 be the ‘awakener‘ of political participation, in addition to being the ultimate legitimizer 84 of the representative democratic model (Robles and Córdoba 2019).“ (p. 2) But if the internet is the awakener of political participation, this relates to participatory democratic theories rather than representative democratic theories. The latter do not want citizens to participate in the political discourse, they should only observe politics and vote every few years.
Another weakness in the theory section is that it is completely unrelated to the research strand on the role of emotions on social media (to name just a few examples: Blassning et al., 2021; Eberl et al. 2017; Jost, Maurer & Haßler, 2020). It seems like the authors haven’t done any literature research in this field. It is necessary to relate their study with the existing research in the field that they want to contribute to. In relation to that, it is noticeable how short the list of references is.
Moreover, the authors take a purely pessimistic perspective on social media for information consumption but ignore that algorithmic curation on social media can also have positive aspects, for example incidental news exposure (Kümpel, 2019) and increasing exposure diversity (Bodó et al., 2019). The authors write furthermore that the danger of echo chambers leads to social media enhancing a discourse that doesn’t fit the ideals of the public debate (I assume they mean the ideals of deliberative theories). However, I think a much more important reason for that is that communication on social media is so short – for example, only 280 signs on Twitter – and written communication is always prone to misunderstandings. Social media are simply not the right place for high-quality political debates. They also state that only “little research has questioned the existence of the ideal digital public sphere as envisaged by Habermas“ (p. 2). This is not true, there are tons of studies which have shown exactly that political debates on the Internet are far from matching Habermasian ideal standards, and this is now rather state of the art (e.g., Rußmann, 2012).
Also methodologically, I have several serious doubt. First of all, I don’t know what a “methodological emotivist approach“ (p. 4) is, and this is no where explained in the paper. The term “motives“ is neither defined in the paper even though it seems to be central to the analysis. Then, I think most of the indicators are not suited to measure what the authors state to investigate. Related to the wrong understanding of deliberative theories (see above) is a signifikant weakness in the operationalization. What the study measures is if the tweets contain “content“. However, it is unclear what the authors mean with that. But content can be everything, I cannot think of any tweets without content. Also purely expressing emotions is a kind of content. A tweet without content would be completely empty. Maybe the authors mean “political content“ but they don’t write that. This must be clarified. And still, providing political content is not suitable as an indicator for checking if tweets fulfill the requirements of deliberative theories since these require much more than just “content“. Containing anything does not make a tweet matching an “argumentative format“, whatever the authors mean with that.
The authors differentiated between tweets positive and negative towards Trump. But most probably not all tweets were unambigious in this respect. What happened with the ambigious ones, how were these coded? I neither understand how the use of a hashtag can be an indicator of arguments being present in a tweet. A hashtag is not an argument, so what is the connection here?
Concerning the labeling of the 3000 tweets, was a reliability test conducted? If yes, the results need to be reported. If not, this raises doubts on the reliability of the coding.
When it comes to the sample, it is unclear to me if Trump’s own tweets were included in it. If yes, this might be a reason for the differences the authors find between 2016 and 2020. It may be that Trump himself as president has poisened the discussion climate on Twitter with his highly problematic tweets. It is also unclear if the sample also contained retweets.
When it comes to the interpretation of the results, the authors seem to evaluate it negative if people became more critical towards Trump on Twitter over time. I don’t understand why this is a problem. Is the core idea of the authors that people shouldn’t change their mind? If people become more critical towards Trump who violated a lot of deliberative standards and other norms, isn’t that rather an indicator of people being interested in a deliberative discourse?
The authors close their paper with writing: “ Authors should discuss the results and how they can be interpreted from the per-489 spective of previous studies and of the working hypotheses. The findings and their impli-490 cations should be discussed in the broadest context possible. Future research directions 491 may also be highlighted.“ (p. 13) This is interesting since this is exactly what the authors themselves don’t do. The discussion is a pure summary of the findings without any discussion or interpretation of the findings, and both a discussion of the limitations and the implications of the study are missing.
No where in the paper is explained why the US are a relevant case for investigating the role of emotions on Twitter in political communication. This must be justified, and the authors must provide background information about their case (e.g., how many people in the US used Twitter at the time of investigation?). The US are an exception in many respects, they are a very polarized country, and Trump is a very special person, which makes the findings of the study not transferrable to other contexts. So why do we then need this study? Why do the authors think that the quality of the discourse was better in 2016 compared to 2020? This must be explained. I also wonder how comparable both elections are when Trump was once challenger and once incumbent. This could change how people (and himself) behaved on Twitter.
Minor things
- A headline should be included before the theory section. At the moment, there is no differentiation between the introduction and the theory section.
- “To verify this hypothesis“ (p. 11) – if there is a hypothesis, why is it not derived in the theory section but rather first presented in the results section? Must be moved to the theory section.
References
Blassnig, S., Udris, L., Staender, A., & Vogler, D. (2021). Popularity on Facebook during election campaigns: An analysis of issues and emotions in parties’ online communication. International Journal of Communication, 15, 21.
Bodó, B., Helberger, N., Eskens, S. & Möller, J. (2019). Interested in Diversity. The role of user attitudes, algorithmic feedback loops, and policy in news personalization. Digital Journalism, 7(2), 206-229.
Eberl, J.-M., Tolochko, P., Greussing, E., Song, H., Lind, F., Heidenreich, T., . . . Boomgaarden, H. (2017). Emotional reactions on Austrian parties’ Facebook pages during the 2017 Austrian Parliamentary election. University of Vienna. Retrieved from https://compcommlab.univie.ac.at/fileadmin/user_upload/p_compcommlab/CCL_Reactions_Report.pdf
Kümpel, A. S. (2019). The issue takes it all? Incidental news exposure and news engagement on Facebook. Digital Journalism, 7(2), 165–186.
Jost, P., Maurer, M., & Hassler, J. (2020). Populism Fuels Love and Anger: The Impact of Message Features on Users’ Reactions on Facebook. International Journal Of Communication, 14, 22. https://ijoc.org/index.php/ijoc/article/view/13400
Rußmann, U. (2012). Online Political Discourse in Facebook: An Analysis of Political Campaign Communication in Austria. Zeitschrift für Politikberatung 3, 115-125.
Author Response
Dear Reviewer,
Thank you very much for all your commets. We have tryed to answer all of them and we very much appreciate your time. Attached you will find a document where you will see our answers.
Yours faithfully.

Reviewer 3 Report
Comments and Suggestions for Authors
The paper under review examines the emotional and argumentative content of Tweets focusing on Donald Trump or his political rivals in 2016 and 2020. The authors make conclusions regarding the trends in positive versus negative emotions aimed at Trump across the two time periods and the (lack of) substantive content beyond the emotional expressions.
The data used and questions explored are of high interest. The literature review is relevant, but I do not think it is especially thorough. There are certainly entire additional areas of relevant research regarding emotional appeals and rational appeals in politics and their relative use and effectiveness that could be included in both the introduction and the discussion. Improving the thoroughness of the literature review would improve the quality of structure, academic soundness, and engagement with recent scholarship aspects of the scoring criteria.
The paper includes numerous helpful and well-crafted visualizations of the study data. However, Figure 4 is labeled in Spanish, and clarification is needed regarding Figure 8. In the main body of the paper the authors state in regard to Figure 8 that they "decided to filter the tweets with hashtags". The figure is labeled "Distribution of Tweets without Hashtags". Does this mean the tweets with hashtags were filtered out?
Overall, I think the paper shows promise but could benefit from a moderate level of revision to improve engagement with additional relevant research and clarify some aspects of reporting results.
Comments on the Quality of English LanguageN/A
Author Response
Dear Reviewer,
Thank you very much for all your comments. We have tried to answer all of them and we very much appreciate your time. You will find our answers below.
The paper under review examines the emotional and argumentative content of Tweets focusing on Donald Trump or his political rivals in 2016 and 2020. The authors make conclusions regarding the trends in positive versus negative emotions aimed at Trump across the two time periods and the (lack of) substantive content beyond the emotional expressions.
The data used and questions explored are of high interest.
Theoretical framework and discussion:
The literature review is relevant, but I do not think it is especially thorough. There are certainly entire additional areas of relevant research regarding emotional appeals and rational appeals in politics and their relative use and effectiveness that could be included in both the introduction and the discussion. Improving the thoroughness of the literature review would improve the quality of structure, academic soundness, and engagement with recent scholarship aspects of the scoring criteria.
Thank you very much for your comment. We have already introduced new references to improve the article.
Evans, H.K., Gervais, B.T., Russell, A. (2022). Getting Good and Mad: Exploring the Use of Anger on Twitter by Female Candidates in 2020. In: Foreman, S.D., Godwin, M.L., Wilson, W.C. (eds) The Roads to Congress 2020. Palgrave Macmillan, Cham. https://doi.org/10.1007/978-3-030-82521-8_4
Gelman, J., Lloyd Wilson, S., & Sanhueza Petrarca, C. (2021). Mixing messages: How candidates vary in their use of Twitter. Journal of Information Technology & Politics, 18(1), 101–115. https://doi.org/10.1080/19331681.2020.1814929
Losada, J.C., Robles, J.M., Benito, R.M., Caballero, R. 2022. Love and Hate During Political Campaigns in Social Networks. In: Benito, R.M., Cherifi, C., Cherifi, H., Moro, E., Rocha, L.M., Sales-Pardo, M. (eds) Complex Networks & Their Applications X. Complex Networks 2021. Studies in Computational Intelligence, vol 1073. Springer, Cham. https://doi.org/10.1007/978-3-030-93413-2_6
Results:
The paper includes numerous helpful and well-crafted visualizations of the study data. However, Figure 4 is labeled in Spanish, and clarification is needed regarding Figure 8. In the main body of the paper the authors state in regard to Figure 8 that they "decided to filter the tweets with hashtags". The figure is labeled "Distribution of Tweets without Hashtags". Does this mean the tweets with hashtags were filtered out?
Thank you very much for pointing out the mistake of Figure 4. In the new version, we have corrected it. Thank you also for the observation that refers to figure 8. The text that appears at the bottom of the graph is correct. On the other hand, we have corrected the term “filter” to “filter out” in the text that refers to it.
Yours faithfully.
Reviewer 4 Report
Comments and Suggestions for Authors
The article offers an original research approach to the communication content of social networking platforms, creating a useful model. The phenomenon of emotivism is appropriately conceptualized. The authors devote most of their attention to explaining their method and analyzing the results of its application, providing rich data material that contributes to the development of digital research methods. The results are adequately described, offering a brief discussion of the potential development directions of this research method. The article is clearly structured, offering a short and purposeful literature analysis chapter. Authors demonstrate good theoretical knowledge and skills to create an original digital research model.
I propose to consider the following additions and improvements:
- Although the greatest attention was paid to the understanding of emotions in political communication, the article's literature review does not mention Shantal Moffe's contribution to the conceptualisation of an affective public sphere; I recommend to evaluate it at least in the discussion part.
- The methods section describes Twitter data collected before the US elections in 2016 and 2020, but the situation of political communication is not explained. Such an addition would enable readers to understand the context of the research presented in the article and to assess the significance of the study;
- I recommend including in the article data and references to other Twitter communication studies, which analyze communication during the 2016 and 2020 pre-election period in the USA.
Comments on the Quality of English LanguageNo specific comments, just figure 4 is not fully translated to English.
Author Response
Dear Reviewer,
Thank you very much for all your comments. We have tried to answer all of them and we very much appreciate your time. You will find our answers below.
The article offers an original research approach to the communication content of social networking platforms, creating a useful model. The phenomenon of emotivism is appropriately conceptualized.
Thank you for your comments. Following your recommendations, we have just added some more conceptualization of emotivism.
Methods:
The authors devote most of their attention to explaining their method and analyzing the results of its application, providing rich data material that contributes to the development of digital research methods.
Results:
The results are adequately described, offering a brief discussion of the potential development directions of this research method.
Structure and theoretical frame:
The article is clearly structured, offering a short and purposeful literature analysis chapter. Authors demonstrate good theoretical knowledge and skills to create an original digital research model.
I propose to consider the following additions and improvements:
- Although the greatest attention was paid to the understanding of emotions in political communication, the article's literature review does not mention Shantal Moffe's contribution to the conceptualisation of an affective public sphere; I recommend to evaluate it at least in the discussion part.
Following your recommendation, we have introduced the proposed reference in the Discussion section. Thank you very much.
- The methods section describes Twitter data collected before the US elections in 2016 and 2020, but the situation of political communication is not explained. Such an addition would enable readers to understand the context of the research presented in the article and to assess the significance of the study;
It is not common to include in academic articles a report on the political situation of the countries under study. Among many other reasons, due to issues related to the extension of this type of academic work and, on the other hand, because this type of evaluations have a high degree of speculation and subjectivity. Likewise, for the method and results of this study, the situational factor of political communication is not a determining factor.
- I recommend including in the article data and references to other Twitter communication studies, which analyze communication during the 2016 and 2020 pre-election period in the USA.
Following your recommendation, we have introduced more references. Thank you very much.
Comments on the Quality of English Language
No specific comments, just figure 4 is not fully translated to English.
Thank you, we have corrected the mistake.
Yours faithfully
Reviewer 5 Report
Comments and Suggestions for Authors
This article analyzes tweets related and referring to Trump during two US election campaigns (2016 and 2020), aiming to identify the supposed sentiment of these messages and whether they contain any form of content, argumentation, or reasoning. The assumption is that tweets lacking arguments convey simple emotions and lack substantive content. Considering this primary objective, the descriptive results and comparisons between the two analyzed samples make this work relevant and interesting in the field of political communication in digital environments, aligning well with the Special Issue to which it has been submitted. However, there are aspects that could be improved, and several limitations that need to be acknowledged and clarified in the manuscript.
The following are comments and suggestions for improvement, highlighting possible limitations:
Firstly, the introduction, study justification, contextual framework, and state of the art seem adequate in quality and extension, allthough more and updated references could have been used.
Moving on directly to the method, several limitations need to be noted. Initially, it is recommended to clarify that the Twitter API is no longer open or available for research. Formally, the term "scales" seems misused in reference to each of the studied dimensions (sentiment and presence of content), and it would be more appropriate to use terms like "variables" or "measures".
Concerning the detection models, conventional shallow learning algorithms were used for their generation. These models are nearly obsolete and are rarely used because they lack the reliability and precision achievable with deep learning algorithms or BERT models. Additionally, the dataset used for training the models was generated manually labeling only 3000 tweets, which appears to be a small sample for training two models to automatically detect sentiment and the presence of argumentation in messages. In addition, to generate the final training corpus, in the case of the model for sentiment analysis, those messages that had been labeled as neutral were removed (after manual labeling), resulting in a final sample of 2250 tweets –1574 used for training and 676 as test set–. And in the case of the content model, all those tweets in which there were hashtags were eliminated (after manual labeling), reducing the sample to a total of 2,206 messages –1,458 used for training and 748 as test set– (also it is not clear why the percentage of the sample used for training and testing is different for each of the models). Considering this, the final datasets seem too small to generate reliable models (in fact, in the case of the content model, the metrics do not seem to show an adequate level of accuracy), and the lack of external validation after model generation raises concerns about their validity and reliability.
Furthermore, what is understood and defined as sentiment analysis actually seems to be misunderstood and misused. The approach does not seem entirely correct, since what is being defined in the study as a sentiment analysis, in reality it would not be so, since the criteria for identifying a positive or negative sentiment are confused. In a sentiment analysis what we try to detect is exactly that, the latent or underlying feeling that exists in the messages, regardless of the support or rejection emitted by those messages. And, in this case, what is being evaluated is whether the message, precisely, emits some type of support or rejection of Trump (or of his political opponents), understanding this as positive or negative. In this sense, it must be understood that many messages of support or rejection of a specific actor can be expressed indistinctly through positive feelings (such as joy, hope, or excitement) or negative feelings (such as sadness, disappointment, or anger). So, this would actually be an analysis of support for Trump, but not sentiment analysis. Furthermore, detailed information is missing on the criteria used in manual labeling (how and by whom it was carried out, whether a single coder or several coders was used...), as well as on the model adjustment parameters: e.g. What coefficient had to be detected to identify a message as positive or negative? Were neutral messages not detected? Did the model automatically discard them?
Along these lines, it would also be worth analyzing and/or reflecting on whether neutral tweets are tweets with content. Neutral tweets are supposed to be messages in which there is no latent sentiment (they could be simple informational messages), and, therefore, it would be normal for the majority of them to be messages with content. However, in the detection results no figures are given for neutral messages (despite the fact that a high number of neutral messages were found in the previous manual labeling).
On the other hand, the title of Figure 4, "Confusion matrix for the sentiment model," seems to be dedicated to the "Confusion matrix of the content model."
At the end of the results, it is indicated that a statistical test has been run to check whether there were significant differences between the number of messages with content identified in the 2016 elections and in the 2020 elections, but it is not specified which specific test has been performed. Furthermore, no data is given to support this test. In this same sense, considering that one of the main objectives of the study was to compare both samples, that of 2016 with that of 2020, more statistical analyzes are missing that would allow us to corroborate whether there really are significant differences, for example, in the number of negative/neutral/positive tweets found, or in support of Trump…
Finally, while the discussion and conclusions are interesting, a more in-depth discussion and final reflections referencing previous research in this line of research are missing. And most importantly, there is a lack of a final section or paragraph in which all the limitations of the work indicated here are made explicit, while at the same time future lines of research are offered.
With all these observations, it is considered that the article is interesting, appropriate and could be published in the Special Issue, as long as the suggested changes and improvements are addressed.
Comments on the Quality of English LanguageThe text is well written in English, although a proofreading in recommended and some minor corrections should be addressed.
Author Response

(The authors gave the same response as above.)

Round 2
Reviewer 2 Report
Comments and Suggestions for Authors
The revisions made by the authors and their response letter were not able to dispel my concerns, unfortunately. My impression is that the authors made only superficial changes to the paper which were easy to implement rather than dealing intensively with the points I made. I don’t expect authors to include all suggestions made by the reviewers but if they don’t do, I want to be convinced by arguments in the response letter that it is the right decision not to implement the suggested changes. The authors were not able to convince me in this case. Moreover, I have the feeling that the authors did not really understand how fundamental my concerns were. I can’t see that the authors really have taken my comments serious and have put much effort in using the reviews for improving the quality of their paper.
One of my main concerns was that the authors have a superficial understanding of deliberative democratic theories and that their study does not really fit in this strand of research. The authors' reaction to that is that they explain in the response letter that their goal is not to investigate deliberative democratic theories. However, when referring to terms such as "deliberation on the internet", "the deliberative idea", and "deliberative public sphere", they relate their study to this theoretical tradition of democratic theories. And then it must be expected that they have a more profound understanding of this theoretical approach rather than using these terms as buzzwords. I don't mean that the authors should investigate deliberative theories in their entirety. However, if stating that they are interested in the deliberativeness of the discourse on Twitter, then I expect from a high-quality, social scientific study that it (a) contains a profound explanation of that theory (without which the study lacks a theoretical foundation) and (b) that the measuring instruments are suited to investigate the deliberativeness of the discourse on Twitter (which is not the case right now). This is the norm in social scientific research. If the authors don't want to familiarize themselves with deliberative theories, they should not create the impression by means of buzzwords that their study would contribute to this strand of research.
The authors state in their response letter that «Although we mention deliberation, our objective is not to address the study of the deliberative democratic theories. This is a very extensive field. The aim of our paper is much more specific and empirical.» I understand that the authors can’t and don’t want to study deliberative democratic theories in their entirety, and this is neither what I wanted them to do. However, a good empirical study needs a profound theoretical basis, it needs to define the concepts it investigates, and the measuring instruments need to fit to these concepts. And what I wanted to say is that this is what is missing. Only with a profound theoretical foundation, an empirical social scientific study like this one can make a significant contribution.
Despite the authors’ explanation in the response letter, I still don’t understand what they mean with «a methodological emotivist approach». Besides parts of the explanation being in Spanish (and me not speaking Spanish), saying that the approach is not ontological (which I’m not sure makes sense) does not help to understand what they mean with «methodological emotivist approach». What exactly does «methodological emotivist» comprise, besides not being ontological?
I still don't understand how tweets were coded that contained both positive and negative sentiments as the authors only differentiate between either positive or negative. The authors did not answer this question in their response letter, unfortunately. I still think a binary classification oversimplifies the reality of tweets.
The authors write: «The manual labeling that we refer to in chapter 2.1 was made, tested and agreed upon by experts in social sciences and in the US electoral system.» Ok, but what I actually meant was an inter-coder reliability test, using a coefficient such as Krippendorff’s alpha or Brennan’s and Prediger’s kappa to check if several human coders arrive at the same result when coding the same materials. Such a test is an important means of quality assurance in manual standardized content analyses and a norm in social scientific content analysis. When 3000 tweets are coded by human coders following certain rules, this is a manual standardized content analysis.
«In any case, our study is not designed to answer these types of questions.» Yes, of course the study is not suited to answer such rather normative questions. But still, the discussion should discuss potential implications of the study to show its societal relevance. This is missing from the discussion.
Concerning comment 12: No, this was not a misunderstanding. I understood well that the word “authors“ referred to the scientific community. However, what I wanted to point out was: When the authors obviously think that it’s important to do the things they rise in the discussion for future research – interpreting their results against the background of previous studies and discussing the implications in a broader context. If this is important (and they seem to think this is important, otherwise they would not write that future research should do that), why don’t they do it themselves in their study? Why should only other authors discuss the findings of their study but not themselves? And again, parts of the answer are in Spanish.
Concerning the selection of the US as a case, I am familiar with the overly strong focus of research on the US which at the same time means that many other regions are strongly neglected by research so far. However, saying that many other studies have focused on the US is not a good enough reason for also focusing on the US. That others have investigated a certain country does not necessarily mean that it is a relevant case. Given that the US are exceptional in so many respects, there is rather an urgent need for research NOT focusing on the US.
“Regarding the number of US citizens using Twitter during the period studied we do not consider that relevant information to interpret our results.“ Not for interpreting the results but for justifying why the case is relevant.
Author Response
Thank you very much for your commnets. Attached you will find our answers. We hope we have solved your main concenrs.

Reviewer 3 Report
Comments and Suggestions for Authors
I can recommend publication of the revised manuscript.
Author Response
Thank you very much for your revision.